Nigeria; mental health; mhGAP; implementation; primary care

**Corresponding author:**
Theddeus Iheanacho;
Email: theddeus.iheanacho@yale.edu

# Integrating mental health into primary care in Nigeria: Implementation outcomes and clinical impact of the *HAPPINESS* intervention

Theddeus Iheanacho[1] , Casey Chu[2], Chinyere M. Aguocha[3], Emeka Nwefoh[3,4] and Charles Dike[1]

[1]Department of Psychiatry, Yale School of Medicine, New Haven, CT, USA; [2]Department of Health Policy and Management, Yale School of Public Health, New Haven, CT, USA; [3]Department of Medicine, Imo State University, Owerri, Nigeria and [4]CBM International, Abuja, Nigeria

## Abstract

**Background:** The Health Action for Psychiatric Problems In Nigeria including Epilepsy and SubstanceS (*HAPPINESS*) intervention is a multicomponent, community-based, mobile technology-supported intervention that integrates mental health into primary health care centers in Nigeria using the World Health Organization's Mental Health Gap Action Programme-Intervention Guide (mhGAP-IG). This study evaluates its implementation and patient-level clinical impact using a quasi-experimental design (single cohort with pre- and post-measures). **Findings:** The HAPPINESS intervention implementation demonstrated high feasibility with 84% adoption rate (% of participating primary health centers that completed its roll out) and 81% fidelity (% of clinicians who completed required intervention components according to the protocol). Retention rate in care at 12 months was 86%. Among patients with complete clinical records analyzed ($n = 178$), there was a statistically significant reduction in 9-item Patient Health Questionnaire scores from baseline ($Md = 9.5$) to 6 months ($Md = 3.0$) post-intervention ($z = 80.5$, $p < 0.001$), with a large effect size ($r = 0.8$) and statistically significant reduction in Brief Psychiatric Rating Scale scores from baseline ($Md = 36.0$) to 6 months ($Md = 17.0$) post-intervention ($z = 128.5$, $p < 0.001$), with a large effect size ($r = 0.9$). **Implications:** Mobile technology-enhanced, mhGAP-IG-based efforts to scale-up mental health services in Nigeria are feasible and effective.

## Impact statement

This study of the Health Action for Psychiatric Problems In Nigeria including Epilepsy and SubstanceS (HAPPINESS) Intervention, which aims to integrate mental health into primary care settings, adds to the growing evidence of the feasibility, preliminary evidence of possible effectiveness, and impact of the WHO's Mental Health Gap Action Programme-Intervention Guide (mhGAP-IG)-based efforts to scale-up mental health services in Nigeria. The HAPPINESS intervention led to a significant reduction in 9-item Patient Health Questionnaire scores among patients from baseline ($Md = 9.5$) to 6 months ($Md = 3.0$) post-intervention ($z = 80.5$, $p < 0.001$), with a large effect size ($r = 0.8$). Additionally, it specifically highlights the potential role of mobile technology and telemedicine in supporting capacity building in mental health in Nigeria and making evidence-based interventions accessible at the community level. A critical component of task-sharing and collaborative care model is clinical supervision and support by mental health specialists. The HAPPINESS intervention's approach to supervision and clinical support for trained healthcare workers can be the model for a national scale-up. It is important to conduct a larger, cluster-randomized trial of the HAPPINESS intervention to further test its definitive effectiveness, sustainability, scalability and cost-effectiveness.

## Social media summary

The *HAPPINESS Intervention* increases access to care by integrating mental health care into primary health care through task sharing.

## Introduction

With only 250 psychiatrists for 200 million people, one psychologist per 2 million people, one psychiatric nurse per 103,000 people, and no dedicated government funding for community

mental health centers (Esan et al., 2014; APN, 2023), Nigeria exemplifies the severe lack of mental health specialists and services that is prominent in many low- and middle-income countries (LMICs) (Kakuma et al., 2011; WHO, 2020). The reasons for this paucity of mental health specialists and services are multifactorial, but an effective approach to mitigating the shortage is training nonmental health professionals to deliver packages of mental health care in a collaborative, stepped-care, task-sharing approach (Patel et al., 2010; Padmanathan and De Silva, 2013; Raviola et al., 2019). These trained nonspecialists are typically supported both virtually and remotely by mental health specialists through regular consultation, referral, supervision, and continuing education as has been demonstrated in the Project Extension for Community Healthcare Outcomes (ECHO) initiative across multiple countries (Serhal et al., 2018; Sockalingam et al., 2018).

In line with this approach, Nigeria's strategic policy on mental health articulated in 1991 is to integrate mental health care into primary, secondary and tertiary care settings with the responsibility of implementation delegated to local governments (Nigeria FMH, 1991). The policy was revised in 2013 to accommodate logistic challenges of integrating mental health care within the primary health system and to make specific recommendations for primary, secondary, and tertiary care (Federal Ministry of Health, 2013). These recommendations included: (a) at the primary care level, having psychotropic medications from the *essential drug list*, strengthening community outreach, awareness, rehabilitation and referral/consultation processes; (b) at the secondary care level, setting up inpatient and outpatient mental health services at all hospitals and strengthening inter-sectoral governance structures to manage mental, neurological and substance (MNS) use disorder services and (c) at the tertiary care level, supporting secondary care services with mental health expertise. Furthermore, the revised policy encouraged the use of public-private partnerships to further innovate mental health service delivery. Finally, it recommended adequate training, retraining and continuing professional development activities on screening, identification, treatment and referral of MNS disorders at all levels of care for all direct care medical personnel.

Despite these long-standing, well-articulated policy goals, mental health integration into general health settings in Nigeria has not been fully implemented due to factors related to funding, training and supervision of trained nonmental health specialists (Saraceno et al., 2007; Petersen et al., 2019). Of the few states in Nigeria (Benue, Lagos, Osun and Ogun) that have successfully implemented state-wide mental health training for primary health care workers, the results are favorable, showing improvement in knowledge, enhanced capacity to treat common mental disorders and improvement in patients' clinical symptoms (Adebowale et al., 2014; Gureje et al., 2015; Adewuya et al., 2019; Ryan et al., 2020).

The World Health Organization's Mental Health Gap Action Programme-Intervention Guide (mhGAP-IG), which supports this approach to scale-up treatment and services for MNS disorders in LMICs (Keynejad et al., 2021), has been contextualized and piloted in Nigeria and utilized in programme implementation in these states (WHO, 2010; Abdulmalik et al., 2013; Keynejad et al., 2018). Core components of the mhGAP-IG approach are primary healthcare center (PHC)-level screening, identification, and treatment of MNS disorders, training for nonmental health specialists, specialist supportive supervision for the trainees and strengthening referral pathways between trained health care workers and specialists (WHO, 2015). However, some implementation research questions have not been addressed. For instance, few studies have focused on fidelity and sustainability of the mhGAP-IG protocols and other manualized processes for assessment, treatment and referral delivered by trained, nonspecialist workers (Gureje et al., 2015; Davies and Lund, 2017; Onofa et al., 2023). There is also a need to identify how best to

efficiently access specialist support, supervision, consultation and referral for trained clinicians providing mental health care in nonmental health specialist centers (Naslund et al., 2021). Furthermore, there is limited data on the rate of adoption of mhGAP-IG-based collaborative care models in primary, secondary and tertiary care centers in Nigeria (Timothy et al., 2018; Faregh et al., 2019).

This study evaluates the implementation and patient-level clinical impact of the mhGAP-IG-based Health Action for Psychiatric Problems including Epilepsy and SubstanceS (HAPPINESS) intervention pilot integrating mental health into PHCs in Imo State Nigeria from October 2019 to September 2021. We hypothesized that the HAPPINESS intervention would be delivered with fidelity by the trainees and that the intervention would improve patients' symptoms as measured by the 9-item Patient Health Questionnaire (PHQ-9) and Brief Psychiatric Rating Scale (BPRS).

## Methods

### Study design

This study used a quasi-experimental design (single cohort with pre- and post- measures) to assess changes in psychiatric symptoms using patients' scores on the PHQ-9 and BPRS, which were documented by PHC clinicians at baseline and 6 months post-intervention for all patients treated through the HAPPINESS intervention. PHQ-9 is a multipurpose instrument for screening, diagnosing, monitoring, and measuring the severity of depression (Levis et al., 2019). BPRS is a rating scale used to measure psychiatric symptoms such as psychosis, depression and anxiety (Overall and Gorham, 1988). Both the PHQ-9 and BPRS have been validated and widely used for research in Nigeria (Adewuya et al., 2006; Gureje et al., 2019). Although the mhGAP-IG training includes modules on other mental disorders, substance use disorders and epilepsy, for the purpose of this pilot study, we chose to use the PHQ-9 and BPRS to measure symptoms of depression, anxiety and psychosis as these are the most common presentations in the community. Measures of HAPPINESS intervention feasibility (*site recruitment, training completion*); adoption (*rate of implementation*) and fidelity (*delivery of screening, assessment, treatment, referral and consultation according to the intervention protocol*) were assessed through administrative data review completed at 12 months post HAPPINESS intervention roll out at each primary care site (Appendix 1 of the Supplementary Material).

### Setting

Imo State has a population of about 4 million people (National Population Commission, 2020) with 527 PHCs staffed by 453 nurses, 76 community health officers and 864 community health extension workers spread across the State's urban and rural areas. Data obtained during the needs assessment indicated that each center had a part-time or full-time covering physician. Additionally, health services at the PHCs are free at point of care although access to essential medications is limited and patients often pay for medications out-of-pocket. During planning meetings, 3 additional PHCs were added to the initial 10 PHCs from 5 local government areas of the state that were selected for the pilot study. The PHCs were selected based on the availability of staff in the clinics and to ensure equitable geopolitical representation of the three senatorial districts of the state as recommended by the stakeholders. A total of 25 nurses (*10 with prior in-service training in mental health and 15 with no prior in-service training in mental health*) from 13 participating PHCs were invited for the HAPPINESS intervention mhGAP-IG 5-day training and 1-day 6-month refresher training, including

training on the use of mobile apps for continuing medical education, specialist consultation and referral. All HAPPINESS intervention clinical services including screening, assessment, treatment and care coordination were delivered in the community PHCs. As is generally the case in Imo State, most of the patients live within 1–2.5 km from the participating PHCs (Charles-Akalonu et al., 2021), making the HAPPINESS intervention accessible.

### Study population and sample

The target population for the study was all patients attending PHCs for routine medical care and services in the participating 13 PHCs. Medical records reviewed during the planning phase of the project showed that the PHCs served a total of 2,756 adult patients in the preceding 12 months. These patients came for services like antenatal care, immunization and general medical care. The study sample included adults who screened positive for MNS disorders based on clinical assessment, PHQ-9 and BPRS scores.

### Health care workers training

Details of the HAPPINESS intervention components, including *Training, Refresher Training, Clinical Practice, Support Supervision, Community Engagement and the Drug Revolving Fund* (Appendix 2 of the Supplementary Material), have been previously published and summarized in Figure 1 (Chu et al., 2022). In summary, for the mhGAP-IG training, we chose the following modules: *Essential Care and Practice, Depression, Psychoses/Mania, Epilepsy and Substance Use Disorders* and made relevant adaptations based on local needs and context. The training was conducted in English, the official language in Nigeria, and delivered in person, facilitated by the mhGAP-IG master and trained trainers. It was an 8-h per day, 5-day training (with scheduled breaks) with didactics, group workshops and role-plays. Post-training certification was based on 100% attendance to all training activities, a score of at least 90% on the post-training knowledge test, and a "pass" grade on the *observed practice session* as scored by trainers according to pre-set competency rubrics (patient assessment, patient education, diagnosis and treatment).

### Post-training support supervision, consultation and referral

Supportive supervision was conducted in person (when feasible) and virtually (WhatsApp® video call/regular mobile phone call) with the covering psychiatrist completing at least once a month,

1-h check-in with the HAPPINESS intervention-trained nurses in the PHCs. Consultation and referral between the PHCs nurses and supporting psychiatrists were conducted in person when feasible or using one of three mobile applications: WhatsApp® Messaging, voice or video call, "one2one" and/or VULA health app (Gloster et al., 2021; one2one), depending on internet availability. The one2one and VULA mobile telehealth applications were specifically developed for LMICs in Africa to improve access to health care, particularly in community settings with few numbers of specialists. For the HAPPINESS intervention pilot, primary health nurses were trained to use both apps to deliver clinical care, consult and make specialist referrals when feasible and appropriate. Continuing medical education was conducted at least once a month using WhatsApp® Group Messaging or Zoom® Meetings. Detailed data collection on the staff, patients' and specialists' utilization of mobile technology in the HAPPINESS intervention is ongoing.

### Clinical tool kit and standard operating procedure

The HAPPINESS intervention Clinical Tool Kit and Standard Operating Procedure outlined screening, assessment and treatment protocol. It also included clinical documentation, data entry, appointment management, referral and consultation processes (Appendices 1 and 2 of the Supplementary Material).

### Data collection

Using the HAPPINESS project implementation checklists, guided by Proctor et al.'s implementation outcomes framework (Proctor et al., 2011) and chart review, data were abstracted from training reports, PHC monthly reports (prepared by the nurses as part of their routine PHC reports), clinic registers of new patients and notes on treatment and follow-ups (Figure 2). Patients' diagnoses and scores on PHQ-9 and BPRS questionnaires at baseline and 6 months post-intervention were documented and collated using a project data sheet.

### Data analysis for clinical outcomes

We conducted the Mann–Whitney $U$ and Kruskal–Wallis tests, which are nonparametric statistical tests used to compare medians of independent samples, to compare differences between sex, marital status, presence of anxiety and/or depression and nurse type for baseline or post-intervention PHQ-9 and BPRS scores, separately.

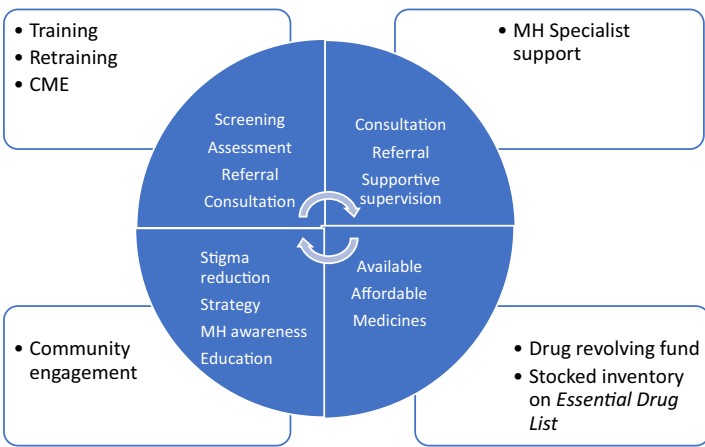

**Figure 1.** Key components of the Health Action for Psychiatric Problems in Nigeria including Epilepsy and SubstanceS (HAPPINESS) intervention.

| Implementation metrics | Level of analysis | Data collection tools and techniques |
|---|---|---|
| **Adoption**: (1) % of primary health workers who completed HAPPINESS training, (2) % of PHC with HAPPINESS intervention set up (3) % of monthly reports submitted to the data collation team | Primary Health Care facility | • Using a checklist, data abstraction from<br>  ○ Training report,<br>  ○ Facility report<br>  ○ Monthly data collation report |
| **Feasibility**: (1) % of eligible patients screened for MNS disorders the past 12 months (2) % of eligible patients treated through the HAPPINESS intervention in the past 12 months (3) % of eligible patients retained in treatment in the past 12 months. | Primary Health Care facility | • Using a checklist, data abstraction from<br>  ○ Screening registry<br>  ○ Treatment registry |
| **Fidelity**: % of health workers adhering to the HAPPINESS Protocols:  (*Assessments, Treatments, Follow ups, Documentation*) | Individual clinician | • Using a checklist, chart review |

**Figure 2.** Implementation data collection metrics and tools.

We also conducted the Wilcoxon signed-rank test, which is a nonparametric test used to compare medians of dependent samples to compare baseline or post-intervention median scores for the PHQ-9 and BPRS overall and separately per category. We used the R statistical software (Thomas et al., 2013) to conduct all analyses and calculated associated effect sizes, which are the rank-biserial correlation coefficient (*r*) for Wilcoxon signed-rank, the standardized difference between two groups' ranks (dividing the test statistic by its maximum value, which is the product of the Ns for the two groups) for Mann–Whitney *U* (Conroy, 2012) and eta-squared ($\eta^2$) for Kruskal–Wallis.

The study was approved by Yale University Institutional Review Board (#2000022691) and Imo State University Teaching Hospital's Ethics Committee.

## Results

All invited 25 primary care nurses from 13 primary care clinics, drawn from 5 out of the 27 *Local Government Areas* of Imo State, completed the HAPPINESS intervention training in the assessment and management of the following MNS priority conditions: depression, mania/psychosis, epilepsy and substance use disorders. The HAPPINESS intervention was successfully implemented in 11 out of the 13 primary care clinics (84%). Two clinics could not implement the intervention at their sites because the trained nurses at those sites unexpectedly relocated. Out of the expected 132 monthly reports, 6 were missing from one clinic and 126 reports were obtained from the 11 active clinics over the 12-month pilot period (completion rate of 95%), which is a good measure of service penetration.

A total of 1,573 primary care clients were screened for MNS disorders over the 12-month period, with 196 identified and treated for MNS conditions. At 6 months, 178 patients (91%) were engaged in treatment, and at 12 months, 168 patients (86%) were still engaged in treatment.

A random chart review of 44 patient records (4 from each clinic) to assess fidelity to the HAPPINESS intervention protocol using the clinical tool kit checklist (Appendix 1 of the Supplementary Material) showed completion of *Assessments, Treatments, Follow ups and Documentation* in 36 of the records (81%).

Out of the data collected from 196 patients, only 178 had complete records of baseline and post-intervention PHQ-9 and BPRS and could be used in the analysis. Table 1 shows the demographic and other relevant clinical characteristics of the overall sample and the analytical sample. The analytical sample's mean age was 37.5 years, 53.4% of individuals identified as female, and most individuals were single (52.2%) or married (42.7%). Over a quarter of the patients received care from the HAPPINESS intervention mhGAP-trained nurses with prior psychiatric in-service training (*Psych Nurses*) (26.4%), and the rest received care from HAPPINESS intervention mhGAP-trained nurses with no prior psychiatric in-service training (*Non-Psych Nurses*) (73.6%). For psychological diagnoses, 36% of the sample had anxiety and/or depression, 39.3% had psychosis, 21.3% had epilepsy and 7.9% had substance use disorder. Table 2 shows the mean and median overall PHQ-9 and BPRS scores at baseline and at 6 months post-intervention for the original and analytical samples. Overall, the PHQ-9 and BPRS mean and median scores decreased post-intervention, with slightly higher scores in the analytical sample.

The data for PHQ-9 and BPRS were not normally distributed and skewed, as assessed by Shapiro–Wilk tests for pre- (baseline) and 6 months post-intervention (pre-PHQ-9: $W = 0.957$, $p < 0.001$ and post-PHQ-9: $W = 0.827$, $p < 0.001$; pre-BPRS: $W = 0.917$, $p < 0.001$ and post-BPRS: $W = 0.886$, $p < 0.001$), thus our decision to conduct Mann–Whitney *U* (MacFarland et al., 2016) and Kruskal–Wallis tests (Ostertagova et al., 2014).

Results from the Wilcoxon signed-rank test (Table 3) showed a statistically significant reduction in PHQ-9 scores from baseline (*Md* = 9.5) to 6 months (*Md* = 3.0) post-intervention ($z = 80.5$, $p < 0.001$), with a large effect size ($r = 0.8$). It also showed a statistically significant reduction in BPRS scores from baseline (*Md* = 36.0) to 6 months (*Md* = 17.0) post-intervention ($z = 128.5$, $p < 0.001$), with a large effect size ($r = 0.9$). The same was seen for each sex category and nurse type, with significant differences between baseline and post-intervention scores that have a large effect size.

The results of the Mann–Whitney *U* and Kruskal–Wallis tests (Table 4) show that baseline PHQ-9 scores alone were not significantly different by sex, nurse type, or marital status categories.

**Table 1.** Demographic and medical characteristics of the original and analytical samples

| | Original sample (N = 196) | Analytical sample (N = 178) |
|---|---|---|
| **Age** | | |
| Mean (SD) | 37.4 (15.0) | 37.5 (15.2) |
| Median (min, max) | 37.0 (2.00, 80.0) | 36.5 (2.00, 80.0) |
| **Sex** | | |
| Female | 106 (54.1%) | 95 (53.4%) |
| Male | 90 (45.9%) | 83 (46.6%) |
| **Marital status** | | |
| Divorced | 1 (0.5%) | 1 (0.6%) |
| Married | 85 (43.4%) | 76 (42.7%) |
| Separated | 6 (3.1%) | 5 (2.8%) |
| Single | 101 (51.5%) | 93 (52.2%) |
| Widowed | 3 (1.5%) | 3 (1.7%) |
| **Nurse type** | | |
| Non-psych nurses | 135 (68.9%) | 131 (73.6%) |
| Psych nurses | 61 (31.1%) | 47 (26.4%) |
| **Psychiatric diagnoses** | | |
| Depression | 65 (33.2%) | 64 (36.0%) |
| Anxiety | 36 (18.4%) | 35 (19.7%) |
| Anxiety and depression | 36 (18.4%) | 35 (19.7%) |
| Dementia | 7 (3.6%) | 7 (3.9%) |
| Psychosis | 83 (42.3%) | 70 (39.3%) |
| Substance use disorder | 16 (8.2%) | 14 (7.9%) |
| Epilepsy | 40 (20.4%) | 38 (21.3%) |
| Presence of general medical diagnosis* | 8 (4.1%) | 7 (3.9%) |

*Note*: The number of individuals and percent of the sample are reported in parentheses.
*Medical diagnoses included malaria, hypertension, diabetes and chronic pain.

**Table 2.** Overall means and averages of PHQ-9 and BPRS scores

| | Original sample (N = 196) | Analytical sample (N = 178) |
|---|---|---|
| **Baseline PHQ-9** | | |
| Mean (SD) | 9.64 (6.51) | 10.5 (6.14) |
| Median (IQR) | 9.0 (4.50, 17.50) | 9.5 (5.25, 17.75) |
| **6 month (post) PHQ-9** | | |
| Mean (SD) | 3.76 (2.82) | 4.14 (2.68) |
| Median (IQR) | 3.0 (1.50, 9.50) | 3.0 (2.00, 9.50) |
| **Baseline BPRS** | | |
| Mean (SD) | 38.4 (18.30) | 39.5 (18.30) |
| Median (IQR) | 35.0 (17.5, 68.5) | 36.0 (21.5, 68.5) |
| **6 month (post) BPRS** | | |
| Mean (SD) | 17.8 (9.15) | 18.3 (9.28) |
| Median (IQR) | 16.5 (8.25, 36.25) | 17.0 (11.50, 36.50) |

It was, however, significantly different between those with and without a depression or anxiety diagnosis, with a large effect size ($U = 4,471$, $p < 0.05$, effect size = 0.613). Baseline BPRS scores were not significant for any of the categories.

On the other hand, post-intervention PHQ-9 scores alone were not significantly different by sex or depression or anxiety diagnosis. However, they were significantly different by nurse type with a moderate effect size ($U = 2,462$, $p < 0.05$, effect size = 0.400) and marital status with a small effect size ($\chi^2(4) = 10$, $p < 0.05$, $\eta^2 = 0.0346$). Post-intervention BPRS scores were not significantly different by sex, depression or anxiety diagnosis or marital status. However, they were significantly different by nurse type with a large effect size ($U = 4,323$, $p < 0.05$, effect size = 0.702).

## Discussion

Results from the pilot implementation of the HAPPINESS intervention showed a high rate of adoption (84%), penetration (service report completion rate of 95%) and fidelity (81%) as well as patient engagement (91%) and retention in care at 12 months (86%). These promising results add to the initial findings of high acceptability and feasibility of the HAPPINESS intervention (Chu et al., 2022). They are also in keeping with other studies of mhGAP-based integration of mental health into primary care in Nigeria and other LMICs (Gureje et al., 2015; Petersen et al., 2019). These implementation outcomes also provide needed data and basis for further development and testing of the HAPPINESS intervention in a larger cluster randomized controlled trial (RCT) to determine its definitive effectiveness compared to usual care and potential for sustainment and scale-up nationally.

The HAPPINESS intervention also led to significant improvements in patients' BPRS and PHQ-9 scores at 6 months post-intervention across all demographics and diagnoses. This finding adds to the growing evidence that collaborative, task-shifting, stepped-care model of mental health care delivery using the WHO's mhGAP-IG can be an effective approach to scaling up mental health in Nigeria and increasing access to evidence-based care in the community for people with common mental disorders (Adewuya et al., 2019; Gureje et al., 2019). The HAPPINESS intervention is particularly relevant as its key components: remote specialists' supervision, consultation and referral as well continuing medical education for the trainees, utilized telemedicine and mobile technology. The option of remote, virtual supervision, consultation and continuing education allowed the inclusion of psychiatrists in Nigeria and Nigerian psychiatrists in the diaspora. This model of remote expert clinical support and continuing education has been well-studied in the Project ECHO model in high-income countries and LMICs (Hager et al., 2018; Sockalingam et al., 2018). It offers a pathway for many psychiatrists and mental health specialists originally from LMICs currently in the diaspora to potentially support clinical education, capacity building and mental health care delivery in their home countries.

When looking at improvement in patients' scores on the PHQ-9 and BPRS across time, patients who were treated by nurses with prior mental health experience and those who were treated by nurses without prior mental health experience had significantly different baseline and post-intervention scores, showing that both groups of patients benefited from the intervention over time. However, further

**Table 3.** Wilcoxon signed-rank test results for baseline and post-intervention, overall and by sex and nurse type

| Category | Baseline score median (IQR) | Post-intervention score median (IQR) | Z-statistic | p-Value | Effect size, r |
|---|---|---|---|---|---|
| Overall | | | | | |
| PHQ-9 | 9.5 (5.25, 17.75) | 3.0 (2.00, 9.50) | 80.5 | 4.94e-26* | 0.826 |
| BPRS | 36.0 (21.5, 68.5) | 17.0 (11.50, 36.50) | 128.5 | 7.36e-30* | 0.852 |
| Female | | | | | |
| PHQ-9 | 10.0 (5.50, 18.00) | 4.0 (2.50, 10.00) | 31.0 | 7.38e-15* | 0.834 |
| BPRS | 38.0 (22.50, 69.50) | 18.0 (12.00, 37.00) | 4.0 | 2.97e-17* | 0.866 |
| Male | | | | | |
| PHQ-9 | 9.0 (5.00, 15.00) | 3.0 (2.00, 7.50) | 16.5 | 1.04e-12* | 0.815 |
| BPRS | 33.0 (20.00, 62.00) | 15.0 (10.50, 35.00) | 61.0 | 3.37e-14* | 0.836 |
| Psychiatric nurse | | | | | |
| PHQ-9 | 7.0 (4.00, 16.00) | 3.0 (2.00, 7.50) | 15.5 | 4.04e-07* | 0.777 |
| BPRS | 36.0 (27.50, 63.50) | 19.0 (14.50, 31.00) | 25.5 | 1.87e-08* | 0.829 |
| Nonpsychiatric nurse | | | | | |
| PHQ-9 | 10.0 (5.50, 18.00) | 4.0 (2.50, 10.00) | 25.5 | 2.41e-20* | 0.840 |
| BPRS | 36.0 (21.50, 68.50) | 14.0 (10.00, 35.00) | 44.0 | 8.36e-23* | 0.858 |

*Note*: Effect sizes close to 1 suggests a strong association (close to 0 suggests a weak or no association).
*Denotes a significant *p*-value at the 0.05 level.

**Table 4.** Mann–Whitney *U* and Kruskal–Wallis test results for baseline and post-intervention scores by sex, nurse type, depression or anxiety diagnosis, and marital status

| | Baseline | | | Post-intervention | | |
|---|---|---|---|---|---|---|
| Category | U statistic | p-Value | Effect size | U statistic | p-Value | Effect size |
| Sex | | | | | | |
| PHQ-9 | 4,336 | 0.25 | 0.550 | 4,336 | 0.20 | 0.550 |
| BPRS | 4,428 | 0.20 | 0.562 | 4,184 | 0.50 | 0.531 |
| Nurse type | | | | | | |
| PHQ-9 | 2,838 | 0.40 | 0.461 | 2,462 | 0.04* | 0.400 |
| BPRS | 2,892 | 0.50 | 0.47 | 4,323 | 4e-05* | 0.702 |
| Marital status | | | | | | |
| | Kruskal–Wallis chi-squared | p-Value | Effect size, $\eta^2$ | Kruskal–Wallis chi-squared | p-Value | Effect size, $\eta^2$ |
| PHQ-9 | 6.159 | 0.1876 | 0.0125 | 10 | 0.03* | 0.0346 |
| BPRS | 5.996 | 0.1995 | 0.0115 | 8 | 0.1 | 0.0219 |

*Note*: All comparisons for two subcategories (sex, nurse type, depression or anxiety) were analyzed using the Mann–Whitney *U* test. Marital status included more than two categories; thus, it was analyzed using the Kruskal–Wallis test. For the Mann–Whitney, effect sizes near 1 suggests a large difference between the groups (close to 0 suggest small or negligible differences). For Kruskal–Wallis, $\eta^2$ between 0.01 and 0.06 indicates small effect), between 0.06 and 0.14 is moderate and greater than 0.14 is large.
*Denotes a significant *p*-value at the 0.05 level.

comparison of post-intervention PHQ-9 and BPRS scores between the two groups of patients show that the treating nurses with prior mental health experience had a substantial impact on the scores. The baseline and post-intervention medians, along with the results of the Mann–Whitney *U* (Tables 2 and 4) show that there is a high probability that a randomly chosen patient who was treated by a

nurse with prior mental health experience will have a lower (and more favorable) post-intervention score than a randomly chosen patient who did not receive care from a nurse with prior mental health experience. The magnitude of this effect was larger for BPRS than PHQ-9. Altogether, this means that although patients treated by either nurse type improved over time, post-intervention scores were lower among individuals who received care from nurses with prior mental health experience (a difference that was not present at baseline). While we are unable to determine exactly the reason for this difference, we speculate that prior experience in treating patients with mental illness, including the use of basic counseling skills gained through specialized training and practice, may have contributed to the nurses' capacity to provide more robust psychosocial support. It also highlights the need for the trainees to continue to practice the skills they learned during this pilot project to solidify their learning and further enhance their clinical competence (Kazantzis et al., 2010).

As expected, baseline PHQ-9 scores significantly differed between those with and without a depression/anxiety diagnosis, with a moderate effect size. This difference was not seen in post-intervention scores, indicating that improvement in symptoms following treatment for those with depression or anxiety led to reduced PHQ-9 scores, and the psychosocial support delivered as part of the HAPPINESS intervention benefitted even those without depression or anxiety. This supports the need to include psychosocial support and counseling as part of a basic package of care for people with MNS use disorders, irrespective of specific diagnoses. Post-intervention PHQ-9 scores were also significantly different by marital status, but the small effect size indicates that only a small proportion of the variance in the PHQ-9 scores can be explained by marital status.

## Limitations

First, this was a quasi-experimental study with baseline and 6 months post-intervention assessments. Therefore, there is a risk of confounding, which limits the confidence in establishing causality compared to an RCT. However, the study provides robust data to support a larger cluster RCT that can support measures to mitigate against confounders.

Second, due to the non-normal nature of the data, we could not conduct parametric tests (i.e., a mixed ANOVA) to compare the main and interaction effects between time (baseline and post-intervention) and different categories or factors of interest (e.g., gender).

Third, although we attempted to include representative primary care centers from the three geopolitical zones of the state, this was a relatively small patient sample size compared to the general population of the state. Thus, we are unable to confidently generalize the findings.

Fourth, although there were significant improvements in PHQ-9 scores among patients who received treatment, we were unable to determine if the improvement in symptoms of anxiety, depression or psychosis were due to psychosocial support plus medication or psychosocial support alone. In depression, for example, the general recommendation is "watchful waiting" and psychoeducation for PHQ-9 scores of 5–9 and counseling, follow-up and/or pharmacotherapy for PHQ-9 scores of 10–14 and above (Iglesias-González et al., 2018); however, the data entry in our pilot sites did not differentiate those who received psychoeducation plus medication or psychoeducation/counseling alone based on their scores. In our subsequent planned, larger RCT, we plan to gather more details to enable the evaluation of the effects of psychosocial support versus the addition of medication management.

Finally, due to budget constraints, we were only able to assess limited implementation outcomes, including measures of adoption, fidelity and retention in treatment. As we plan and propose a larger, more robustly funded study, we will include additional implementation outcome measures focused on sustainability, cost-effectiveness and potential for a national scale-up using a mixed-method approach.

## Conclusion

This pilot implementation study shows that it is feasible to integrate mental health into primary care with fidelity and a high adoption rate. It also shows that primary care nurses trained with mhGAP-IG and provided specialist clinical support physically and remotely using mobile technology can lead to improvement in patients' symptoms with a high rate of retention and engagement in care.

**Open peer review.** To view the open peer review materials for this article, please visit http://doi.org/10.1017/gmh.2024.4.

**Supplementary material.** The supplementary material for this article can be found at http://doi.org/10.1017/gmh.2024.4.

**Data availability statement.** The datasets used and/or analyzed during the current study are available from the corresponding author on reasonable request.

**Acknowledgments.** The authors are grateful to their volunteer project support staff especially Ms. Obianuju Ukaigwe for her dedication and diligence in coordinating data collection from all participating primary health care centers. The authors would also like to acknowledge the support provided Imo State Primary Care Development Agency, Imo State University Teaching Hospital, Imo State Ministry of Health and HHC Foundation.

**Author contribution.** TI, CA, EN and CD were responsible for the design of the HAPPINESS intervention and training. TI and CA also contributed to data collection. CC and TI conducted data collation and data analysis. All authors provided significant input and approved the final manuscript.

**Financial support.** Funding for this study was provided by the Yale Institute of Global Health Hecht Faculty Award.

**Competing interest.** The authors declare none.

**Ethics statement.** Ethical approval was granted by Yale University's Institutional Review Board and Imo State Teaching Hospital's ethics committee. Both institutions assessed that the research posed no more than minimal risk of harm to subjects and involves no procedures other than administrative, clinical chart review and data abstraction.

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
