## [Reviewer Report]

To the Editor, Global Mental Health

September 3, 2023

Dear Editor, 

We wish to submit an original research article entitled “Integrating mental health into primary care in Nigeria: Implementation outcomes and clinical impact of the HAPPINESS intervention” for consideration by Global Mental Health journal. 

We confirm that this manuscript has not been published elsewhere, nor is it currently under consideration by another journal.

In this paper, we evaluate implementation outcomes of the HAPPINESS intervention pilot that utilized the World Health Organization’s Mental Health Gap Action Programme Intervention Guide (mhGAP-IG) to train primary care teams to provide treatment for mental, neurological and substance use disorders in the community. Mobile technology and telehealth apps were used for clinical support, consultation, and referral.

We believe that this manuscript is appropriate for publication by Global Mental Health because it transparently highlights the challenges and successes of a mental health innovation and system strengthening strategy in a low and middle-income country. Our study comes at a time in Nigeria when a new mental health legislation has been passed with emphasis on integration of mental health care into primary care. Thus, our findings show a potential model for scaling up mental health care across the country. 

All authors have approved the manuscript and agree with its submission. We also have no competing interests to disclose.

Thank you for your consideration of this manuscript. 

Sincerely

Theddeus Iheanacho, M.D.

Associate Professor of Psychiatry, Yale University School of Medicine

Visiting Professor of Psychiatry,

Imo State University

Director, Yale Global Mental Health Program

Department of Psychiatry,

Yale University School of Medicine

Director, Global Mental Health Promotion Program

Center for Methods in Implementation and Prevention Science (CMIPS)

Yale School of Public Health

Chair of Psychiatry and Director of Behavioral Health Services

Trinity Health of New England/Saint Francis Hospital

Email: theddeus.iheanacho@yale.edu

Tel: +14752029380

https://medicine.yale.edu/psychiatry/happiness/